# Seroprevalence of *Chlamydia trachomatis*, herpes simplex 2, Epstein-Barr virus, hepatitis C and associated factors among a cohort of men ages 18–70 years from three countries

**Shams Rahman**[1], **Deanna Wathington**[1], **Tim Waterboer**[2], **Michael Pawlita**[2], **Luisa L. Villa**[3], **Eduardo Lazcano-Ponce**[4], **Martina Willhauck-Fleckenstein**[2], **Nicole Brenner**[2], **Anna R. Giuliano**[5]*

1 Bethune-Cookman University, Daytona Beach, Florida, United States of America, 2 German Cancer Research Center (DKFZ), Heidelberg, Germany, 3 School of Medicine, University of São Paulo, São Paulo, Brazil, 4 Instituto Nacional de Salud Publica, Cuernavaca, Mexico, 5 Center for Infection Research in Cancer, Moffitt Cancer Center, Tampa, Florida, United States of America

* anna.giuliano@moffitt.org

**Data Availability Statement:** All relevant data required to replicate the study's findings are within

## Abstract

### Objectives

To estimate the seroprevalence of *Chlamydia trachomatis* (CT), herpes simplex type-2 (HSV2), hepatitis C (HCV), Epstein-Barr virus (EBV) and nine human papilloma virus (HPV) types, and investigated factors associated with the seropositivity among men from three countries (Brazil, Mexico and U.S).

### Methods

Archived serum specimens collected at enrollment for n = 600 men were tested for antibodies against CT, HSV2, HCV, EBV, and 9-valent HPV vaccine types (6/11/16/18/31/33/45/52/58) using multiplex serologic assays. Socio-demographic, lifestyle and sexual behavior data at enrollment were collected through a questionnaire.

### Results

Overall, 39.3% of the men were seropositive for CT, 25.4% for HSV2, 1.3% for HCV, 97.3% for EBV, 14.0% for at least one of the seven oncogenic HPV (types: 16/18/31/33/45/52/58), and 17.4% for HPV 6/11. In the unadjusted models, age, race, smoking, sexual behavior variables, and seropositivity for high-risk HPV were significantly associated with the seropositivity for CT. In multivariable analyses, self-reported black race, higher numbers of lifetime female/male sexual partners, current smoking, and seropositivity to high-risk HPV were significantly associated with increased odds of CT seropositivity. Odds of HSV2 seroprevalence were elevated among older men and those seropositive for high risk HPV.

the paper. There are restrictions on sharing the raw datasets publicly. However, in compliance with the IRB protocols and patient privacy laws, they can be provided upon written request to Julie Rathwell, Sr. Research Project Specialist, Moffitt Cancer Center and Research Institute (Email: Julie. Rathwell@Moffitt.org; Phone: 813-745-6471).

**Funding:** This work was funded by a grant from the National Cancer Institute, National Institutes of Health (R03: CA176743, PI: Anna R. Giuliano), and the HIM Study cohort infrastructure was supported through a grant from the National Cancer Institute, National Institutes of Health (R01: CA098803, PI: Anna R. Giuliano).

**Competing interests:** The authors have declared that no competing interest exist.

## Conclusion

Exposure to STIs is common among men. Prevention and screening programs should target high-risk groups to reduce the disease burden among men, and to interrupt the disease transmission to sexual partners.

## Introduction

Sexually transmitted infections (STIs) are a major public health problem causing serious morbidity and mortality worldwide. STIs have a direct impact on reproductive health, newborn health, pregnancy complications, and cancer [1] and facilitate transmission of human immunodeficiency virus (HIV), and human papillomavirus (HPV) [2, 3]. STIs can be transmitted from one person to another through sexual contact including vaginal and anal intercourse, oral-genital contact, sex-toys and kissing [4]. Each year an estimated 499 million new cases of STIs occur worldwide [5]. In the United States approximately 20 million new cases of STIs occur every year [6]. There are more than 30 different types of STIs [7] with the most common and important types being, *Chlamydia trachomatis* (CT), herpes simplex type 2 (HSV2), hepatitis C virus (HCV), Epstein-Barr virus (EBV), human papillomavirus (HPV), syphilis, gonorrhea and human immunodeficiency virus (HIV). Not all cases of STI are reported due to the subclinical course of some cases and the reluctance of some patients to visit a healthcare provider to seek treatment for clinical cases.

CT, a gram-negative bacterium, is one of the most commonly diagnosed STIs worldwide with approximately 131 million new cases each year [8]. In 2012, 1.43 million new cases of CT were diagnosed in the U.S. [9]. HSV2, a double-stranded DNA virus that causes genital herpes [10] is particularly common with an estimated 417 million prevalent cases aged 15–49 years worldwide in 2012 [11]. The most recent data on HSV2 were published in March of 2010 based on the National Health and Nutrition Examination Survey (NHANES) 2005–2008 data. The seroprevalence of HSV2 was 16.2% in those aged 14–49 years in the U.S. [12]. EBV is a double stranded DNA virus that is detected in nearly all populations of the world, acquired early in life, EBV is mainly transmitted through bodily fluids, primarily saliva. EBV can also be transmitted through blood and semen, and organ transplantation [13]. EBV is best known as a causative agent for infectious mononucleosis; however, recent evidence classifies it an STI, as well [8]. EBV seroprevalence in the U.S. is estimated at approximately 70% [14]. HCV is a single-stranded RNA virus that can causes cirrhosis and hepatocellular carcinoma among those chronically infected. Hepatocellular carcinoma in the U.S. is on the rise due to undiagnosed HCV infections [15]. Approximately 3% of the world's population and approximately 1% of the U.S. population is positive for anti-HCV antibodies [16–18].

Seroprevalence estimates and data on the risk factors for STIs are needed to inform populations at greatest risk of disease and for the implementation of public health prevention interventions. Although data on STI prevalence among men in the U.S. are available from large national surveys such as NHANES, data from low- and middle-income countries (LIMC) are limited. In addition, the NHANES has a narrowly defined range for age (14–59 years) provides prevalence for general population, and typically is under-representative of high-risk groups, such as men who sex with men (MSM) and others. Previously we reported the seroprevalence and associated factors of cutaneous HPV and 9-valent HPV vaccine types [19, 20] In this study we extend our analysis to estimate the seroprevalence of CT, HSV2, HCV, and EBV, and investigated factors associated with the seroprevalence among men from three countries: Brazil, Mexico, and the United States.

## Methods

### Study population

Study participants included a sub-cohort of 600 men obtained using simple random sampling (SRS) method from the parent cohort (*the HPV Infection in Men (HIM) Study)*. The *HIM Study* uses a prospective cohort design to examine the natural history of HPV infections among men in three countries. Both participants in the sub-cohort [19, 20] and the full *HIM Study* cohort [21, 22] have been described previously in detail. Briefly, the parent cohort recruited over 4000 men from three study sites (São Paulo, Brazil, Morelos, Mexico, and South Florida, U.S). The *HIM Study* eligibility criteria included: (1) ages 18 to 70 years; (2) resident of one of the three study sites; (3) no current or prior diagnosis of anal or penile cancers; (4) no present or past history of genital or anal warts; (5) no current diagnosis, symptoms or treatment for any sexually transmitted infection; (6) not a participant of an HPV vaccination study; (7) no history of HIV or AIDS; (8) no history of being imprison, homeless or on drug use during the past 6 months of the study screening visit; (9) willingness to participate in 6 months apart 10 follow up visits for 4 years (10) does not have plans to relocate during the study period. The *HIM Study* participants were interviewed and examined every six months for a median of four years. At baseline and each study visit, participants completed a computer-assisted self-interviewed questionnaire, provided blood and urine specimens, and underwent a clinical examination. The current study analyzes a simple random sample of (n = 600) subjects obtained from a total of 3,695 eligible the *HIM Study* participants. Only questionnaire data and serum specimen collected at the *HIM* study baseline were examined for the (n = 600) sub-set. Due to limited resources at our disposable, performing serology testing for the entire parent cohort was neither feasible nor cost-effective. For the sub-set selection, we used the SRS robust approach to account for sampling error by not including everyone in the parent cohort. Written approval of the study protocol and informed consent were obtained from the Institutional Review Boards of the University of South Florida (Tampa, FL, USA), the Ludwig Institute for Cancer Research (Sao Paulo, Brazil), the Centro de Referencia e Treinamento em Doencas Sexualmente Transmissiveis e AIDS (Sao Paulo, Brazil), and the Instituto Nacional de Salud Publica de Mexico (Cuernavaca, Mexico). Each patient read and signed the informed consent form before their participation in the study.

### Specimens and data collection

At the baseline visit, *HIM Study* participants provided detailed information on sociodemographic characteristics, smoking habits, recent alcohol consumption, medical history, and sexual behaviors. Archived baseline serum specimens from participants were tested for antibodies against *Chlamydia trachomatis* major outer membrane (MOMP) and translocating actin-recruiting (Tarp) proteins; herpes simplex virus type 2 envelope glycoprotein 2 (mgG-2 unique); hepatitis C virus core and non-structural (NS3) antigens; and Epstein Barr virus zebra protein, viral capsid antigen (VCA p18) and early-antigen D (EA-D). Seroreactivity to L1 major capsid proteins of the 9-valent HPV vaccine types (6, 11, 16, 18, 31, 33, 45, 52 and 58) was also tested [23]. Antibodies were detected using glutathione S-transferase (GST) capture ELISA in combination with fluorescent bead technology that has the ability to detects type-specific antibodies against each infection tested. The GST-based multiplex serologic essay used in this study has been described in greater details previously [23, 24].

### Statistical analysis

The sub-cohort and full cohort were comparable with respect to socio-demographic factors listed in the Table 2 (see S1 Table for results from the comparisons.) CT, HSV2, HCV, and

EBV seroprevalence was defined as the proportion of men who tested positive for CT, HSV2, HCV or EBV, respectively. Seropositivity to high-risk HPV category was defined as the proportion of men who tested positive for at least one of seven oncogenic types (i.e. 16, 18, 31, 33, 45, 52 and 58). Low-risk HPV category was defined as the proportion of men who tested positive for non-oncogenic types 6 or 11. Two subjects with inconclusive serology results were excluded from all analyses resulting in a final sample size of 598 men.

Baseline sociodemographic and behavioral characteristics of participants were compared between seropositive and seronegative men for CT and HSV2, using Chi-square, and when one or more cells had an expected frequency of less than 5, then the Fisher exact test was used (Table 2). Seroprevalence estimates were calculated for each STI and compared by country, age and HPV sero-status, using Chi-square tests. To examine associations between individual STI seropositivity and potential risk factors, logistic regression was used and odds ratios (ORs) and their 95% confidence intervals (CI) were estimated. Factors listed in Table 2 were considered for inclusion in the multivariable logistic regression models. Variables were selected through bivariate analysis and a backward stepwise elimination process with a significance level of $p \leq 0.2$. Country and age were forced into the models due to the study design. To assess the individual contribution of each variable retained in the model, the likelihood ratio test at $p < 0.1$ was performed. Final multivariable models were estimated only for CT and HSV2. Because only 2.7% of the sample was seronegative for EBV, and only 1.3% of the sample was seropositive for HCV, we did not have sufficient power to estimated multivariable models for these two infections. All analyses were performed in SAS 9.3.

## Results

CT, HSV2, HCV, EBV and HPV seroprevalence estimates for the sub-set (n = 600) at the baseline are presented in Table 1. Overall, 39.3% of the men were seropositive for CT, 25.4% for HSV2, 1.3% for HCV, 97.3% for EBV, 14% for at least one of the seven oncogenic HPV types (16, 18, 31, 33, 45, 52, 58) and 17.4% were seropositive for HPV 6 or 11. Except for EBV and low-risk HPV (6/11), seropositivity to other STIs did not differ significantly by country. Except for low-risk HPV, seropositivity to other STIs showed a significant increasing trend with age

**Table 1. Seroprevalence of *Chlamydia trachomatis*, herpes simplex type 2, hepatitis C, and Epstein-Barr virus by country, age and high-risk HPV antibody status.**

| Infection | | Country | | | | Age group (years) | | | | High-risk HPV[c] | | |
| --- | --- | --- | --- | --- | --- | --- | --- | --- | --- | --- | --- | --- |
| | | | | | | | | | | Serostatus | | |
| | Overall | U.S. | Brazil | Mexico | P[b] | (18–30) | (31–44) | (45–70) | P[b] | Positive | Negative | P[b] |
| | (N = 598)[a] | (n = 184) | (n = 200) | (n = 214) | | (n = 259) | (n = 256) | (n = 83) | | (n = 84) | (n = 514) | |
| | (%) | (%) | (%) | (%) | | (%) | (%) | (%) | | % | % | |
| *Chlamydia trachomatis* | 39.3 | 35.9 | 45.5 | 36.4 | 0.088 | 33.2 | 44.5 | 42.2 | **0.023** | 53.6 | 37.0 | **0.004** |
| Herpes simplex type 2 | 25.4 | 23.4 | 30.0 | 22.9 | 0.188 | 13.5 | 30.1 | 48.2 | **<0.001** | 38.1 | 23.4 | **0.004** |
| Hepatitis C virus | 1.3 | 2.2 | 1.0 | 0.9 | 0.494 | 0.4 | 0.8 | 6.0 | **0.001** | 6.0 | 0.6 | **<0.001** |
| Epstein-Barr virus | 97.3 | 93.5 | 98.5 | 99.5 | **0.001** | 94.6 | 99.2 | 100 | **0.001** | 98.8 | 97.1 | 0.363 |
| High-risk HPV (16, 18, 31,33, 45, 52, 58) | 14.0 | 10.3 | 14.5 | 16.8 | 0.173 | 10.0 | 15.6 | 21.7 | **0.018** | -- | -- | -- |
| Low-risk HPV (6, 11) | 17.4 | 10.3 | 27.5 | 14.0 | **<0.001** | 17.8 | 17.2 | 16.9 | 0.976 | 22.6 | 16.5 | 0.173 |

a. Two subjects with inconclusive test results were excluded from analysis

b. P-value compares seroprevalence for each infection by country, age and HPV status. P-value is from Chi-square test and when one or more cells had an expected frequency <5 then the Fisher exact test was used.

c. Seropositive to at least of these HPV types: (16, 18, 31, 33, 45, 52, 58)

(P<0.05). Also, seropositivity to CT, HSV2, and HCV significantly varied by serostatus to high-risk HPV group (P<0.05). Seropositive men for high-risk HPV were more likely to be seropositive for CT, HSV2 and HCV too.

Participant characteristics by seropositivity to CT and HSV2 are presented in Table 2. Significant differences by age, race, smoking and sexual behavior characteristics were observed for seropositivity to CT (p<0.05). CT seroprevalence was highest among men age 31–44 years, and age 45–73 years, self-reported black race group, current smokers, men who have sex with men (MSM), and men with a increasing number of female/male sex partners. Similarly, significant differences in HSV-2 seroprevalence distribution were noted by age, marital status, and sexual behavior variables (Table 2).

Factors associated with CT and HSV2 seroprevalence are presented in Table 3. In the unadjusted models, age, race, smoking, sexual behavior variables, and high-risk HPV were significantly associated with the CT seroprevalence, and remained significant in multivariable analyses after adjusting for all other variables listed in the table. Compared to white men, black men were more likely to be seropositive for CT (adjusted odds ratio [AOR] 2.36; 95% confidence interval [CI]: 1.33–4.20). Compared to never smokers, current smokers were more likely to be seropositive for CT (AOR 1.65; 95%CI: 1.05–2.58). Compared to men with 1–3 female lifetime sex partners, men with 4–18, and men with ≥19 partners were more likely to be seropositive for CT (AOR 1.68; 95%CI: 1.00–2.82, and AOR 3.12; 95%CI: 1.69–5.76, respectively). Compared to men with 1–3 female lifetime sex partners, men with no reported female lifetime sex partners were also more likely to be seropositive for CT (AOR 2.21; 95%CI: 1.10–4.47); however, the no partner group had reported having male lifetime sex partners instead. Compared to men who reported no male lifetime sex partner, men who reported having ≥2 partners were more likely to be seropositive to CT (AOR 2.28; 95%CI: 1.20–4.33). Seropositivity for CT was also positively associated with seropositivity for high-risk HPV (AOR 1.75; 95%CI: 1.03–2.97). Seropositivity to HSV2 was significantly associated with age, race, marital status, sexual behavior variables, and high-risk HPV in the unadjusted models; however, only associations with age and high-risk HPV remained significant in the multivariable analyses after adjusting for all other factors in the table. Compared to men age 18–30 years, men age 31–44 years, and men age 45 and above were more likely to be seropositive for HSV2 (AOR 3.00; 95%CI: 1.70–5.30 and AOR 6.19; 95%CI: 3.03–12.66, respectively). Men who were seropositive for high-risk HPV were nearly two times more likely to be seropositive for HSV2 (AOR 1.78; 95%CI: 1.01–3.14).

## Discussion

Previously we reported the seroprevalence of cutaneous HPV and mucosal HPV [19, 20]. In this analysis, we report the seroprevalence of commonly diagnosed STIs: *Chlamydia trachomatis* (CT), herpes simplex type 2 (HSV2), hepatitis C virus (HCV), Epstein-Barr virus (EBV), human papillomavirus (HPV), more common among MSM populations. We also examined factors associated with the seropositivity to CT and HSV2. Overall, 39.3% of the men had antibodies to CT, and 25.4% to HSV2. Seropositivity to CT and HSV2 were positively associated with seropositivity to high-risk HPV. It appears that common STIs may transmit together or may share a common source; however, the exact time of transmission cannot be determined in a seroprevalence study.

The seroprevalence of CT (39.3%) in our study was higher than a recently reported seroprevalence of CT (14%) among men aged 16–44 years from a national survey in England [25]. However, the English study reported a seroprevalence estimate of 18.7% for men aged 35–39 years. This difference in CT seroprevalence may in part be explained by the differences

**Table 2. Participant characteristics by serostatus to *Chlamydia trachomatis* and herpes simplex type 2.**

| Characteristic | Serostatus to *Chlamydia* | | | | | Serostatus to herpes simplex 2 | | | | |
|---|---|---|---|---|---|---|---|---|---|---|
| | Negative | | Positive | | Pᵃ | Negative | | Positive | | Pᵃ |
| Country | N | % | N | % | | N | % | N | % | |
| USA | 118 | 64.1 | 66 | 35.9 | 0.088 | 141 | 76.6 | 43 | 23.4 | 0.188 |
| Brazil | 109 | 54.5 | 91 | 45.5 | | 140 | 70.0 | 60 | 30.0 | |
| Mexico | 136 | 63.6 | 78 | 36.4 | | 165 | 77.1 | 49 | 22.9 | |
| **Age, Years** | | | | | | | | | | |
| 18–30 | 173 | 66.8 | 86 | 33.2 | **0.023** | 224 | 86.5 | 35 | 13.5 | **<0.001** |
| 31–44 | 142 | 55.5 | 114 | 44.5 | | 179 | 69.9 | 77 | 30.1 | |
| 45–73 | 48 | 57.8 | 35 | 42.2 | | 43 | 51.8 | 40 | 48.2 | |
| **Race** | | | | | | | | | | |
| White | 167 | 62.1 | 102 | 37.9 | **0.002** | 192 | 71.4 | 77 | 28.6 | 0.079 |
| Black | 32 | 41.0 | 46 | 59.0 | | 54 | 69.2 | 24 | 30.8 | |
| Asian | 13 | 76.5 | 4 | 23.5 | | 16 | 94.1 | 1 | 5.9 | |
| American Indian/Alaska Native | 4 | 44.4 | 5 | 55.6 | | 8 | 88.9 | 1 | 11.1 | |
| Other | 142 | 65.1 | 76 | 34.9 | | 170 | 78.0 | 48 | 22.0 | |
| Missing | 5 | 71.4 | 2 | 28.6 | | 6 | 85.7 | 1 | 14.3 | |
| **Ethnicity** | | | | | | | | | | |
| Hispanic | 177 | 62.8 | 105 | 37.2 | 0.312 | 213 | 75.5 | 69 | 24.5 | 0.624 |
| Non-Hispanic | 179 | 58.7 | 126 | 41.3 | | 225 | 73.8 | 80 | 26.2 | |
| Missing | 7 | 63.6 | 4 | 36.4 | | 8 | 72.7 | 3 | 27.3 | |
| **Education, Years** | | | | | | | | | | |
| 12 or less | 164 | 57.1 | 123 | 42.9 | 0.152 | 213 | 74.2 | 74 | 25.8 | 0.314 |
| 13–15 | 105 | 66.0 | 54 | 34.0 | | 125 | 78.6 | 34 | 21.4 | |
| 16 or more | 94 | 63.1 | 55 | 36.9 | | 106 | 71.1 | 43 | 28.9 | |
| Not reported | 0 | 0 | 3 | 100.0 | | 2 | 66.7 | 1 | 33.3 | |
| **Marital Status** | | | | | | | | | | |
| Single/never married | 159 | 61.4 | 100 | 38.6 | 0.971 | 204 | 78.8 | 55 | 21.2 | **0.013** |
| Married/cohabitating | 168 | 60.4 | 110 | 39.6 | | 205 | 73.7 | 73 | 26.3 | |
| Divorced/separated/widowed | 35 | 60.3 | 23 | 39.7 | | 35 | 60.3 | 23 | 39.7 | |
| Missing | 1 | 33.3 | 2 | 66.7 | | 2 | 66.7 | 1 | 33.3 | |
| **Smoking Status** | | | | | | | | | | |
| Current | 74 | 50.7 | 72 | 49.3 | **0.017** | 110 | 75.3 | 36 | 24.7 | 0.317 |
| Former | 71 | 62.8 | 42 | 37.2 | | 78 | 69.0 | 35 | 31.0 | |
| Never | 218 | 64.3 | 121 | 35.7 | | 258 | 76.1 | 81 | 23.9 | |
| **Alcohol, No. Drinks/Month** | | | | | | | | | | |
| 0 | 87 | 61.7 | 54 | 38.3 | 0.547 | 98 | 69.5 | 43 | 30.5 | 0.161 |
| 1–30 | 174 | 62.4 | 105 | 37.6 | | 213 | 76.3 | 66 | 23.7 | |
| 31–60 | 35 | 60.3 | 23 | 39.7 | | 39 | 67.2 | 19 | 32.8 | |
| 61 or more | 60 | 54.5 | 50 | 45.5 | | 87 | 79.1 | 23 | 20.9 | |
| Missing | 7 | 70.0 | 3 | 30.0 | | 9 | 90.0 | 1 | 10.0 | |
| **Circumcision** | | | | | | | | | | |
| No | 234 | 60.2 | 155 | 39.8 | 0.708 | 290 | 74.6 | 99 | 25.4 | 0.981 |
| Yes | 129 | 61.7 | 80 | 38.3 | | 156 | 74.6 | 53 | 25.4 | |
| **Sexual Orientation** | | | | | | | | | | |
| MSW | 334 | 63.7 | 190 | 36.3 | **0.001** | 401 | 76.5 | 123 | 23.5 | **0.009** |
| MSM | 20 | 39.2 | 31 | 60.8 | | 29 | 56.9 | 22 | 43.1 | |
| MSWM | 7 | 46.7 | 8 | 53.3 | | 11 | 73.3 | 4 | 26.7 | |

*(Continued)*

**Table 2.** (Continued)

| Characteristic | Serostatus to *Chlamydia* | | | | Serostatus to herpes simplex 2 | | | |
|---|---|---|---|---|---|---|---|---|
| | Negative | | Positive | | P[a] | Negative | | Positive | | P[a] |
| Missing | 2 | 25.0 | 6 | 75.0 | | 5 | 62.5 | 3 | 37.5 | |
| **No. of Female LTP** | | | | | | | | | | |
| 0 | 33 | 55.0 | 27 | 45.0 | **0.001** | 47 | 78.3 | 13 | 21.7 | **0.001** |
| 1–3 | 95 | 75.4 | 31 | 24.6 | | 101 | 80.2 | 25 | 19.8 | |
| 4–18 | 158 | 64.2 | 88 | 35.8 | | 195 | 79.3 | 51 | 20.7 | |
| 19 or more | 61 | 48.4 | 65 | 51.6 | | 76 | 60.3 | 50 | 39.7 | |
| Missing | 16 | 40.0 | 24 | 60.0 | | 27 | 67.5 | 13 | 32.5 | |
| **No. of Male LTP** | | | | | | | | | | |
| 0 | 320 | 63.1 | 187 | 36.9 | **0.005** | 384 | 75.7 | 123 | 24.3 | **0.030** |
| 1 | 18 | 64.3 | 10 | 35.7 | | 24 | 85.7 | 4 | 14.3 | |
| 2 or more | 24 | 41.4 | 34 | 58.6 | | 36 | 62.1 | 22 | 37.9 | |
| Missing | 1 | 20.0 | 4 | 80.0 | | 2 | 40.0 | 3 | 60.0 | |
| **Female Sex Patterns in Past 6 Months**[b] | | | | | | | | | | |
| 0 | 67 | 58.3 | 48 | 41.7 | **0.001** | 81 | 70.4 | 34 | 29.6 | 0.209 |
| 1 | 173 | 71.8 | 68 | 28.2 | | 187 | 77.6 | 54 | 22.4 | |
| 2 or more | 81 | 50.0 | 81 | 50.0 | | 115 | 71.0 | 47 | 29.0 | |
| Missing | 9 | 45.0 | 11 | 55.0 | | 16 | 80.0 | 4 | 20.0 | |
| **Male Sex Partners in Past 6 Months**[b] | | | | | | | | | | |
| 0 | 29 | 52.7 | 26 | 47.3 | 0.147 | 43 | 78.2 | 12 | 21.8 | **0.036** |
| 1 | 7 | 53.8 | 6 | 46.2 | | 6 | 46.2 | 7 | 53.8 | |
| 2 or more | 6 | 28.6 | 15 | 71.4 | | 12 | 57.1 | 9 | 42.9 | |
| Missing | 1 | 50.0 | 1 | 50.0 | | 1 | 50.0 | 1 | 50.0 | |

Notes: P = p-value; LTP = lifetime sex partner; MSW = men having sex with women; MSM = men having sex with men; MSMW = men having sex with men and women.

Initial study included 600 men. Two subjects with inconclusive serology results were excluded from all analyses resulting in a final sample size of 598 men.

a. P-value is from chi-square, and when one or more cells had an expected frequency of <5 then the Fisher exact test was used. Missing data were excluded from p-value calculation. Significant p-value is highlighted in bold.

b. Among those reporting ever having a sex partner

population characteristics, risk profile, age group, and the serologic method used in each of the studies. The English study measured antibodies against Pgp3 CT protein, whereas prevalence in our study was based on MOMP and TARP CT proteins. The study population in the English study was much younger (16–44 years), compared to our study (18–70 years).

In a study conducted in Baltimore, U.S. a seroprevalence of 20% among men was reported for CT. This study also consisted of men younger than our study, and seroprevalence was based on antibodies against PgP3 CT protein [26]. The seroprevalence of HSV2 (23.4%) for the U.S. men in our study was also higher than the seroprevalence of (15.7%) reported from the NHANES study for the U.S. population for years 2005–2010 [27]. The NHANES survey participants' age ranged from 14 to 49 years, whereas the age in our study ranged from 18 to 70 years. This conclusion was also somewhat supported when seroprevalence for HSV2 was stratified by age where increasing age was associated with higher levels of seropositivity. The seroprevalence estimates of EBV among U.S. men (93.5%) and HCV (2.2%) in our study were also slightly higher than the seroprevalence of the EBV (70%) and HCV (1.6%) in the U.S. general populations [14, 18].

**Table 3. Factors independently associated with seropositivity for *Chlamydia trachomatis* and herpes simplex type 2.**

| Characteristic | *Chlamydia trachomatis* | | Herpes simplex type 2 | |
|---|---|---|---|---|
| | **Unadjusted** | **Adjusted** | **Unadjusted** | **Adjusted** |
| | **Models (OR; 95%CI)** | **Models (AOR; 95%CI)[a]** | **Models (OR; 95%CI)** | **Models (AOR; 95%CI)[b]** |
| **Country** | | | | |
| USA | *Reference* | *Reference* | *Reference* | *Reference* |
| Brazil | 1.49 (0.99–2.25) | 1.20 (0.72–1.98) | 1.41 (0.89–2.22) | 0.91 (0.51–1.61) |
| Mexico | 1.03 (0.68–1.55) | 1.43 (0.60–3.43) | 0.97 (0.61–1.55) | 1.96 (0.64–6.03) |
| **Age, Years** | | | | |
| 18–30 | *Reference* | *Reference* | *Reference* | *Reference* |
| 31–44 | **1.61 (1.13–2.31)** | 1.27 (0.84–1.93) | **2.75 (1.76–4.30)** | **3.00 (1.70–5.30)** |
| 45–73 | 1.47(0.88–2.43) | 1.06 (0.59–1.92) | **5.95 (3.40–10.41)** | **6.19 (3.03–12.66)** |
| **Race** | | | | |
| White | *Reference* | *Reference* | *Reference* | *Reference* |
| Black | **2.35 (1.41–3.93)** | **2.36 (1.33–4.20)** | 1.11(0.64–1.92) | 1.37 (0.73–2.55) |
| Asian /American Indian/Alaska Native | 0.87 (0.37–2.02) | 0.91 (0.33–2.51) | **0.21(0.05–0.90)** | 0.16 (0.02–1.25) |
| Other | 0.88 (0.60–1.27) | 0.87 (0.37–2.02) | 0.70 (0.46–1.07) | 0.40 (0.14–1.170) |
| **Marital Status** | | | | |
| Single/never married | - - - | - - - | *Reference* | *Reference* |
| Married/cohabitating | - - - | - - - | 1.32 (0.89–1.97) | 0.68 (0.38–1.20) |
| Divorced/separated/widowed | - - - | - - - | **2.44 (1.33–4.46)** | 0.75 (0.35–1.60) |
| **Smoking Status** | | | | |
| Never | Reference | **Reference** | | |
| Current | **1.75 (1.18–2.63)** | **1.65 (1.05–2.58)** | | |
| Former | 1.07 (0.69–1.66) | 1.01 (0.60–1.68) | | |
| **No. of Female LTP** | | | | |
| 0 | **2.51 (1.31–4.81)** | **2.21 (1.10–4.47)** | 2.51 (1.31–4.81) | 0.85 (0.370–1.95) |
| 1–3 | *Reference* | *Reference* | *Reference* | *Reference* |
| 4–18 | **1.71 (1.05–2.76)** | **1.68 (1.00–2.82)** | 1.71 (1.05–2.76) | 0.83 (0.46–1.48) |
| 19 or more | **3.27 (1.91–5.58)** | **3.12 (1.69–5.76)** | 3.27 (1.91–5.58) | 1.76 (0.92–3.36) |
| **No. of Male LTP** | | | | |
| 0 | *Reference* | *Reference* | *Reference* | *Reference* |
| 1 | 0.95 (0.43–2.10) | 0.77 (0.31–1.91) | 0.52 (0.18–1.53) | 0.36 (0.10–1.29) |
| 2 or more | **2.42 (1.39–4.21)** | **2.28 (1.20–4.33)** | 1.91 (1.08–3.37) | 1.18 (0.59–2.37) |
| **High-Risk HPV (16, 18, 33, 45, 52)** | | | | |
| Seronegative | *Reference* | *Reference* | *Reference* | *Reference* |
| Seropositive | **1.97 (1.24–3.13)** | **1.75 (1.03–2.97)** | **2.02 (1.24–3.28)** | **1.78 (1.01–3.14)** |
| **Low-Risk HPV(6, 11)** | | | | |
| Seronegative | *Reference* | *Reference* | *Reference* | *Reference* |
| Seropositive | 1.22 (0.76–1.87) | 0.97 (0.60–1.59) | 1.10 (0.68–1.77) | 1.18 (0.68–2.04) |

OR = odds ratio (unadjusted); AOR = adjusted odds ratio; CI = confidence interval; LTP = lifetime sex partners

a. Adjusted for country, age, race, smoking status, number of female lifetime sex partners and number of male lifetime sex partners

b. Adjusted for country, age, race, marital status, number of female lifetime sex partners and number of male lifetime sex partners

Seropositivity to CT was not associated with marital status in our study (i.e. single/never-married: 38.6%, married/cohabiting: 39.6%, and divorced/widowed/ separated: 39.7%; P-value >0.05). There are several plausible explanations for the lack of this association. It is possible that there is no association between marital status and CT in our study population. Marital

status data were self-reported with a potential for non-differential misclassification. Furthermore, prevalence estimates based on antibodies does not necessarily reflect current infection. Both, changes in marital status, and the accumulation of antibodies accumulation in blood are functions of time, it is difficult to determine when the antibodies detected in the serum of a married person were truly produced. The current study uses a cross-sectional design which in itself has limitation to evaluate cause-and-effect, and temporal sequence. Furthermore, previous studies have also reported mixed results for CT prevalence and marital status association. For example, a study from the U.S. reported a significant protective effect against CT prevalence for married/living with partners when compared to never married (i.e. married: 0.8%, never married: 2.3%, and divorced/widowed/separated: 3%; p-value<0.05) [28]. The U.S. study population included both men and women, and the prevalence was based on urine specimen analysis using the Hologic/Gen-Probe Aptima assay. In contrast, a study from Mexico reported a significant protective effect against CT prevalence for single women when compared to married/living with partners (i.e. married: 16.6%, single: 2.9%, and divorced/widowed/separated: 36.4%; p-value:<0.05) [29]. The Mexico study was conducted in an STD clinic setting, only women were included, and the DNA of CT was detected in endocervical samples through direct fluorescence assay (DFA). A study from Brazil did not find any significant difference for CT prevalence by marital status [30]. The Brazilian study included only men who were recruited from STD clinics, and tested urine for *Chlamydia* DNA using Polymerase Chain Reaction (PCR) method. It can be concluded that the correlation between CT and marital status may depend on the type of study population, study settings (e.g. clinic or population based), whether the study includes men, women or both, the method of infection testing, geographic regions and cultural aspects.

In multivariable analyses, self-reported black race, higher numbers of lifetime female and male sexual partners, and seropositivity to high risk HPV were significantly associated with increased odds of CT seropositivity, consistent with previous reports [31–33]. Current smokers had a significantly increased odds of CT seropositivity. Exposure to tobacco smoke and chemicals increase susceptibility to bacterial infections, including STIs [34]. Seropositivity to HSV2 was also significantly associated with age, race, marital status, sexual behavior variables, and seropositivity to high-risk HPV in the unadjusted models; however, only associations with age and high-risk HPV remained significant in the multivariable analyses. Seropositivity to CT and HSV2 were also positively associated with seropositivity to HPV. Recent evidence suggests that STIs could increase the risk and acquisition and persistence of oncogenic HPV [3]. Findings from this study suggest the concept of co-transmission of STIs and HPV, and perhaps facilitation of transmission of each other, which needs confirmation in longitudinal studies.

Random sample obtained at baseline from a prospective cohort study, one laboratory protocol for serology, comprehensive risk factors data, multi-center, and a broad age range of participants, are some of the strengths of this study. However, certain limitations should be considered when interpreting these results. Although measurement of type specific antibodies against STIs provides direct evidence of exposure to STI, anatomic location, and the time of exposure in serologic studies remain a challenge. Seroprevalence is also subject to the impact of seroconversion rate and antibody decay. Small sample sizes for HCV positive and EBV negative participants limited our ability to assess associated factors. Findings from this study suggest that exposure to STIs is common among men and that age and sexual behavior are key factors of exposure to STI. There are more than 30 different types of STIs [7]. Aside from HPV and hepatitis A and B, there is no effective vaccines available against the other STIs. Prevention programs should target high-risk groups, specifically MSM, to reduce the disease burden among men, and to interrupt the disease transmission to sexual partners.

## Supporting information

**S1 Table. A comparison of the baseline characteristics for the simple random sample and the full parent HIM study cohort.**
(DOCX)

## Acknowledgments

The authors thank the *HIM Study* teams in the United States (Tampa, FL), Brazil (São Paulo), and Mexico (Cuernavaca), as well as the study participants in these three countries.

## Author Contributions

**Conceptualization:** Shams Rahman, Luisa L. Villa, Anna R. Giuliano.

**Data curation:** Shams Rahman.

**Formal analysis:** Shams Rahman, Deanna Wathington.

**Funding acquisition:** Anna R. Giuliano.

**Investigation:** Shams Rahman, Tim Waterboer, Michael Pawlita, Martina Willhauck-Fleckenstein, Nicole Brenner.

**Methodology:** Shams Rahman, Tim Waterboer, Michael Pawlita, Anna R. Giuliano.

**Project administration:** Luisa L. Villa, Eduardo Lazcano-Ponce.

**Resources:** Anna R. Giuliano.

**Software:** Deanna Wathington, Anna R. Giuliano.

**Supervision:** Anna R. Giuliano.

**Validation:** Deanna Wathington, Michael Pawlita.

**Writing – original draft:** Shams Rahman.

**Writing – review & editing:** Shams Rahman, Deanna Wathington, Tim Waterboer, Luisa L. Villa, Eduardo Lazcano-Ponce, Anna R. Giuliano.

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
