## [Decision Letter · Decision Letter 0]

6 Apr 2021

PONE-D-21-05133

Seroprevalence of Chlamydia trachomatis, herpes simplex 2, Epstein-Barr virus, hepatitis c and associated factors among a cohort of men ages 18-70 years from three countries.

PLOS ONE

Dear Dr. Giuliano,

Thank you for submitting your manuscript to PLOS ONE and my apologies that the review process took longer than expected. After careful consideration, we feel that it has merit but does not fully meet PLOS ONE’s publication criteria as it currently stands. Therefore, we invite you to submit a revised version of the manuscript that addresses the points raised during the review process. Please note, that I also served as a second reviewer and my comments are below:

Lines 51-53:    while I am not arguing with statistics you are presenting, I’d probably add a caveat that a lot of STIs may go undiagnosed due to the subclinical course of the disease or reluctance to visit a doctor (quite common at least in the case of EBV or HSV-2)

Line 95:           what was the justification for selecting 600 subjects out of 4000? Also, the setup is quite confusing: the authors mention that the 600 subjects provided serum samples at the baseline, but it is not clear how frequently the samples were collected for the duration of the study? Every 6 months?

Results:           are these results from the baseline? 

Lines 107-111: I think it would be relevant to provide specific information on the tests used in serological analyses

Discussion:      the authors argue that the differences in their observations of CT seropositivity with previously published studies are in part due to the age differences, but their own data (Table 3) show no correlation with age in the adjusted models. Perhaps they need to reconsider their reasoning. In general, the discussion feels rushed and incomplete, I think the authors have a potentially interesting dataset which can be explored better. Most of the findings are somewhat obvious and indicate that increase in STI detection is associated with risky behavior. However, it also poses some unexpected observations, for example, I was surprised to see lack of correlation between marital status and CT seropositivity.  

Overall:           I think the authors present an interesting dataset, but the manuscript needs to be revised at least for clarity. I also think that discussion could be more comprehensive. I'd also like to emphasize point #9 made by the Reviewer #1, which in my view is the most critical.

We look forward to receiving your revised manuscript.

Kind regards,

Edward Gershburg

Academic Editor

PLOS ONE

Journal Requirements:

3. Please upload a copy of Supplement File 1 which you refer to in your text on page 5.

Reviewers' comments:

Reviewer's Responses to Questions

**Comments to the Author**

1. Is the manuscript technically sound, and do the data support the conclusions?

Reviewer #1: Yes

2. Has the statistical analysis been performed appropriately and rigorously? 

Reviewer #1: Yes

3. Have the authors made all data underlying the findings in their manuscript fully available?

Reviewer #1: No

4. Is the manuscript presented in an intelligible fashion and written in standard English?

Reviewer #1: Yes

5. Review Comments to the Author

Reviewer #1: From a statistical and epidemiologic standpoint, this is a straight-forward report of seropositivity rates of a number of STDs in a fairly un-selected sample of men in three countries. The rates of positivity are somewhat interesting. The association between seropositivity and variable such as age, number of sexual partners are of less interest due to their obvious nature. Personally, I would have liked to see the analysis stratified by country because given the different cultures, some of the associations might have varied by country.

I have a number of suggestions to improve the paper, as detailed in my comments to the authors.

1. There is no information in the paper regarding how the participants in the study were identified and recruited. The reader has to go to one of the cited references of the HIM study to find out. This information is crucial to interpreting the seroprevalence results, so authors should make it more readily available. In fact, based on my reading of the cited references, a strength of the study is that the participants were not selected due to some STD condition (which is often the case in these kinds of studies). This information makes the findings more meaningful.

2. The authors refer to using a Chi-Square test to compare the random sub-sample used in this study (n=600) to the full HIM cohort. This is a misuse of a chi-square test. It is based on the common fallacy that the size of the p-value indicates the degree of difference in two groups. That is not the case. A p-value is a function of both the sample size and the size of the difference. For a small sample size a big difference may not be statistically significant, and for a large sample size, a small difference may be statistically significant. I recommend leaving out that sentence.

3. The authors say they used either Fisher’s Exact Test, or Chi-Square Test. They should indicate the criteria they used to decide which to use.

4. Table one has a footnote that says “a. Overall p-value is from Chi-square test”. It is unclear what they mean by that since it appears in an “Overall” column (not the p-value column). What comparison is that referring to?

5. Table 2 can be simplified (no need to show numbers of both negatives, positives, and totals). Simply shown the “n” and the number(%) positive. It would make the table easier to read and compare groups

6. The findings in top sections of Table 3 (country and age) have already appeared in Table 1, so thee rows an be removed.

7. There is no real need to show the unadjusted analysis in Table 3 as this is essentially the same analysis as was presented in Table 2.

8. I think many readers would want to see the show associations between characteristics and positivity, stratified by country.

9. The authors say all relevant data are in the manuscript. That is not exactly true. There is no way to reproduce their multivariable models with the data in the manuscript. And readers with an interest would not be able to produce country-specific results.

6. PLOS authors have the option to publish the peer review history of their article (what does this mean?). If published, this will include your full peer review and any attached files.

Reviewer #1: **Yes: **Laurence Magder

---

## [Author Response · Author response to Decision Letter 0]

18 May 2021

PONE-D-21-05133

Title: Seroprevalence of Chlamydia trachomatis, herpes simplex 2, Epstein-Barr virus, hepatitis c and associated factors among a cohort of men ages 18-70 years from three countries.

..................

Reviewer #1

..................

Lines 51-53: while I am not arguing with statistics you are presenting, I’d probably add a caveat that a lot of STIs may go undiagnosed due to the subclinical course of the disease or reluctance to visit a doctor (quite common at least in the case of EBV or HSV-2)

Thank you for your comment. We have addressed this caveat in the revised manuscript and added the following statement: “Not all cases of STI are reported due to the subclinical course of some cases and the reluctance of some patients to visit a healthcare provider to seek treatment some clinical cases.” Please see page: 3, lines: 55-56

Line 95: what was the justification for selecting 600 subjects out of 4000? Also, the setup is quite confusing: the authors mention that the 600 subjects provided serum samples at the baseline, but it is not clear how frequently the samples were collected for the duration of the study? Every 6 months?

Thank you for your comment. We clarified this confusion in the revised manuscript. In the current manuscript we used data and serum specimens for a sub-set of (n=600) only from the baseline visit. In the parent study (HPV infection in Men or HIM) participants were interviewed and examined every six months for a median of four years. Since the sub-set of n=600 were selected using a SRS sampling methodology, the baseline socio-demographic and sexual behavioral characteristics of the sub-cohort (n=600) and full HIM cohort (>4,000 men) did not statistically significantly differ, which yielded results that would have been obtained if the full cohort was to be examined. We did not have enough resources to perform serology testing for the 4000 subjects of the parent cohort, and it was neither feasible nor cost-effective. Please see page: 4, lines: 99-104

Results: are these results from the baseline? 

Thank you for your comment. The current manuscript only studies baseline seroprevalence and participant characteristics. In the revised version we have clarified this to avoid future confusion. Please see page: 4, lines: 84-85; and Page: 5, lines: 99-101

Lines 107-111: I think it would be relevant to provide specific information on the tests used in serological analyses

Thank you for your comment. In the revised version, we provided information and citations on the specific tests use in the current study. Please see page: 6, lines: 118-121

Discussion: the authors argue that the differences in their observations of CT seropositivity with previously published studies are in part due to the age differences, but their own data (Table 3) show no correlation with age in the adjusted models. Perhaps they need to reconsider their reasoning. 

In general, the discussion feels rushed and incomplete, I think the authors have a potentially interesting dataset which can be explored better. Most of the findings are somewhat obvious and indicate that increase in STI detection is associated with risky behavior. However, it also poses some unexpected observations, for example, I was surprised to see lack of correlation between marital status and CT seropositivity. 

Thank you. In the revised section we addressed some of these concerns and added a new paragraph on lack of correlation between marital status and CT seropositivity. Please refer to the discussion section pages: 14-15, lines: 228-251.

“Seropositivity to CT was not associated with marital status in our study (i.e. single/never-married: 38.6%, married/cohabiting: 39.6%, and divorced/widowed/ separated: 39.7%; P-value >0.05). There are several plausible explanations for the lack of this association. It is possible that there is no association between marital status and CT in our study population. Marital status data were self-reported with a potential for non-differential misclassification. Furthermore, prevalence estimates based on antibodies does not necessarily reflect current infection. Both, changes in marital status, and the accumulation of antibodies accumulation in blood are functions of time, it is difficult to determine when the antibodies detected in the serum of a married person were truly produced. The current study uses a cross-sectional design which in itself has limitation to evaluate cause-and-effect, and temporal sequence. Furthermore, previous studies have also reported mixed results for CT prevalence and marital status association. For example, a study from the U.S. reported a significant protective effect against CT prevalence for married/living with partners when compared to never married (i.e. married: 0.8%, never married: 2.3%, and divorced/widowed/separated: 3%; p-value<0.05) [28]. The U.S. study population included both men and women, and the prevalence was based on urine specimen analysis using the Hologic/Gen-Probe Aptima assay. In contrast, a study from Mexico reported a significant protective effect against CT prevalence for single women when compared to married/living with partners (i.e. married: 16.6%, single: 2.9%, and divorced/widowed/separated: 36.4%; p-value:<0.05) [29]. The Mexico study was conducted in an STD clinic setting, only women were included, and the DNA of CT was detected in endocervical samples through direct fluorescence assay (DFA). A study from Brazil did not find any significant difference for CT prevalence by marital status [30]. The Brazilian study included only men who were recruited from STD clinics, and tested urine for chlamydia DNA using Polymerase Chain Reaction (PCR) method. It can be concluded that the correlation between CT and marital status may depend on the type of study population, study settings (e.g. clinic or population based), whether the study includes men, women or both, the method of infection testing, geographic regions and cultural aspects.”

Please upload a copy of Supplement File 1 which you refer to in your text on page 5.

Thank you. Supplement File 1 was uploaded

....................

Reviewer # 2

....................

1. There is no information in the paper regarding how the participants in the study were identified and recruited. The reader has to go to one of the cited references of the HIM study to find out. This information is crucial to interpreting the seroprevalence results, so authors should make it more readily available. In fact, based on my reading of the cited references, a strength of the study is that the participants were not selected due to some STD condition (which is often the case in these kinds of studies). This information makes the findings more meaningful.

Thank you. We addressed this concern in the revised manuscript. Please see page: 4, lines: 88-96.

2. The authors refer to using a Chi-Square test to compare the random sub-sample used in this study (n=600) to the full HIM cohort. This is a misuse of a chi-square test. It is based on the common fallacy that the size of the p-value indicates the degree of difference in two groups. That is not the case. A p-value is a function of both the sample size and the size of the difference. For a small sample size a big difference may not be statistically significant, and for a large sample size, a small difference may be statistically significant. I recommend leaving out that sentence.

Thank you. We agree with this comment. Although Simple Random Sampling (SRS) is the best sampling strategy to account for sampling error, it does not always guarantee. We removed the statement from the revised manuscript. Please see page: 6, lines: 122-123 (track changes version, the deletion balloon) 

3. The authors say they used either Fisher’s Exact Test, or Chi-Square Test. They should indicate the criteria they used to decide which to use.

Thank you. We addressed this concern in the revised manuscript. Please see page: 6, lines: 131-133, also Tables 1&2 footnotes. 

4. Table one has a footnote that says “a. Overall p-value is from Chi-square test”. It is unclear what they mean by that since it appears in an “Overall” column (not the p-value column). What comparison is that referring to?

Thank you. We apologize there was a typo. The foot notes were switched. We corrected this in the revised manuscript. P-values compares the differences in prevalence by country, age and HPV status. Please see page 6, table 1, footnote a&b.

5. Table 2 can be simplified (no need to show numbers of both negatives, positives, and totals). Simply shown the “n” and the number (%) positive. It would make the table easier to read and compare groups

Thank you. Total was removed; however, we decided to keep the positive and negatives to provide information on the full sample size in each category for the readers. Please see page: 9 table2.

6. The findings in top sections of Table 3 (country and age) have already appeared in Table 1, so thee rows can be removed.

Thank you. We decided to keep these findings to facilitate for readers, for completeness of the table, and to make sure that data in table 3 can be interpreted as self-standing. 

7. There is no real need to show the unadjusted analysis in Table 3 as this is essentially the same analysis as was presented in Table 2.

Thank you very much. This is a valid point. Although the unadjusted odds ratio communicates the same findings as in table 2, we decided to report the un-adjusted odds ratios for two reasons: 1. some readers prefer to look at odds ratios and their corresponding 95% confidence intervals rather than percentages and p-values; 2) presenting the un-adjusted odds ratios alongside with the adjusted odds ratios demonstrates the extent of change in the ORs with adjusting. 

8. I think many readers would want to see the show associations between characteristics and positivity, stratified by country.

Thank you very much. Although stratified analysis is the best approach to control for confounding (by county in this case); however, due to small sample sizes in each group and sub-categories when stratified by country, we lost a lot of power, which restricted our ability to conduct stratified analysis. Instead we restored to regression analysis and used country as an adjusting factor a similar but different approach.

9. The authors say all relevant data are in the manuscript. That is not exactly true. There is no way to reproduce their multivariable models with the data in the manuscript. And readers with an interest would not be able to produce country-specific results. 

Thank you. This is a fair criticism. This criticism is true for many open access and non-open access published studies unless FULL access to RAW data is provided. We will change our initial determination to ‘No’ and mention in the manuscript, that RAW data can be requested via written request to make sure that the IRB protocols are followed, our institutional policies and procedures are not violated, and that the patient’s confidentiality is protected.

Please see page: 17; lines 285-287.

---

## [Editor Report · Decision Letter 1]

27 May 2021

Seroprevalence of Chlamydia trachomatis, herpes simplex 2, Epstein-Barr virus, hepatitis c and associated factors among a cohort of men ages 18-70 years from three countries.

PONE-D-21-05133R1

Dear Dr. Giuliano,

We’re pleased to inform you that your manuscript has been judged scientifically suitable for publication and will be formally accepted for publication once it meets all outstanding technical requirements.

Kind regards,

Edward Gershburg

Academic Editor

PLOS ONE
---

## [Editor Report · Acceptance letter]

14 Jun 2021

PONE-D-21-05133R1 

Seroprevalence of *Chlamydia trachomatis*, herpes simplex 2, Epstein-Barr virus, hepatitis c and associated factors among a cohort of men ages 18-70 years from three countries. 

Dear Dr. Giuliano:

I'm pleased to inform you that your manuscript has been deemed suitable for publication in PLOS ONE. Congratulations! Your manuscript is now with our production department. 

Kind regards, 

on behalf of

Dr. Edward Gershburg 

Academic Editor

PLOS ONE